# The decrotonylase FoSir5 facilitates mitochondrial metabolic state switching in conidial germination of *Fusarium oxysporum*

**Ning Zhang[1†], Limin Song[1†], Yang Xu[1], Xueyuan Pei[2], Ben F Luisi[2], Wenxing Liang[1]***

[1]Shandong Province Key Laboratory of Applied Mycology, College of Plant Health and Medicine, Qingdao Agricultural University, Qingdao, China; [2]Department of Biochemistry, University of Cambridge, Cambridge, United Kingdom

**Abstract** *Fusarium oxysporum* is one of the most important pathogenic fungi with a broad range of plant and animal hosts. The first key step of its infection cycle is conidial germination, but there is limited information available on the molecular events supporting this process. We show here that germination is accompanied by a sharp decrease in expression of FoSir5, an ortholog of the human lysine deacetylase SIRT5. We observe that FoSir5 decrotonylates a subunit of the fungal pyruvate dehydrogenase complex (FoDLAT) at K148, resulting in inhibition of the activity of the complex in mitochondria. Moreover, FoSir5 decrotonylates histone H3K18, leading to a downregulation of transcripts encoding enzymes of aerobic respiration pathways. Thus, the activity of FoSir5 coordinates regulation in different organelles to steer metabolic flux through respiration. As ATP content is positively related to fungal germination, we propose that FoSir5 negatively modulates conidial germination in *F. oxysporum* through its metabolic impact. These findings provide insights into the multifaceted roles of decrotonylation, catalyzed by FoSir5, that support conidial germination in *F. oxysporum*.

**\*For correspondence:**
wliang1@qau.edu.cn

[†]These authors contributed equally to this work

**Competing interest:** The authors declare that no competing interests exist.

## Editor's evaluation

This manuscript is of particular interest to the community of fungal biologists, along with a broader audience studying epigenetic and protein post-translational modifications. The authors combine sophisticated methods to analyze a rather poorly described modification, protein crotonylation mediated by FoSIRT5, and how it affects gene expression/protein activity of metabolically relevant genes/enzymes in Fusarium oxysporum.

## Introduction

*Fusarium oxysporum* is a transkingdom pathogen known to infect more than 100 plant species (*Michielse and Rep, 2009*) and immune-compromised patients (*Nucci and Anaissie, 2007*). Therefore, it is of great importance to gain insight into the molecular processes involved in pathogenesis of this fungus. *F. oxysporum* invades roots and can cause wilt diseases through colonization of xylem tissue. Fungal conidia are the first structures that the host immune system encounters during infection, and conidial germination is a crucial step for *F. oxysporum* infection. The early initiation stage of conidial germination represents a critical point to inhibit fungal growth and counteract pathogenic infection (*Deng et al., 2015*). However, the molecular mechanism by which *F. oxysporum* regulates its germination process remains largely unknown.

Lysine acetylation, one of the most common post-translational modifications (PTMs), is involved in regulation of conidial germination in plant-associated fungi (*Dubey et al., 2019*; *Wang et al., 2018*; *Zhang et al., 2020b*). In recent decades, besides acetylation, numerous other short-chain acylation modifications have been discovered on lysine (K) residues, including crotonylation, malonylation, succinylation, propionylation, glutarylation, and butyrylation (*Chen et al., 2007*; *Hirschey and Zhao, 2015*; *Kim et al., 2006*; *Park et al., 2013*; *Tan et al., 2011*, *Tan et al., 2014*). Among them, lysine crotonylation (Kcr), first identified on histones, is also able to target other proteins involved in various cellular processes (*Wu et al., 2017*; *Xu et al., 2017*). Kcr is recognized by histone-binding 'reader' modules, including AF9 YEATS, YEATS2, MOZ, and DPF2, in a type- and site-specific manner (*Li et al., 2016*; *Xiong et al., 2016*; *Zhao et al., 2016*). Histone crotonylation by p300 has been shown to promote transcription in vitro and manipulating cellular concentration of crotonyl-CoA affects gene expression (*Sabari et al., 2015*). Although a recent proteomic analysis reveals that Kcr is tightly associated with virulence of the necrotrophic fungus *Botrytis cinerea* (*Zhang et al., 2020a*), it is not clear whether this modification is involved in the regulation of fungal germination.

Previous studies have identified mammalian histone deacetylase SIRT1 as being responsible for the removal of crotonylation in the nucleus (*Bao et al., 2014*; *Feldman et al., 2013*). The $NAD^+$-dependent sirtuins (SIRTs) have an expanded repertoire of deacylase activities and display widespread subcellular distributions (*Bell and Guarente, 2011*; *Cen et al., 2011*; *Kanfi et al., 2012*). Three mammalian sirtuins (SIRT3, SIRT4, and SIRT5) localize mostly or exclusively to the mitochondrial matrix, the powerhouse of the cell producing the bulk of cellular ATP through oxidative phosphorylation (*Ryan, 2018*). SIRT3 is considered the major deacetylase of mitochondria (*Lombard et al., 2007*), while SIRT4 mainly functions as a lipoamidase that regulates pyruvate dehydrogenase complex activity (*Mathias et al., 2014*). SIRT5, which possesses poor deacetylase activity (*Du et al., 2011*), preferentially regulates the levels of lysine succinylation, malonylation, and glutarylation, playing multiple roles in regulating different metabolic pathways including glycolysis/gluconeogenesis, fatty acid β-oxidation, oxidative phosphorylation, the urea cycle, and ketogenesis (*Hirschey and Zhao, 2015*; *Park et al., 2013*). However, no information on the regulation of crotonylation in mitochondria is available.

The SIRTs are also present in filamentous fungi and control a variety of cellular processes (*Haigis and Sinclair, 2010*). Seven SIRTs, NST-1 to NST-7, have been identified in *Neurospora crassa*, which mediate telomeric silencing in this fungus (*Smith et al., 2008*). In *Podospora anserina*, deletion of *PaSir2* resulted in a significant reduction of cell life span (*Boivin et al., 2008*). The sirtuins, HstD in *Aspergillus oryzae* and sirtuin A in *Aspergillus nidulans*, control secondary metabolite production (*Itoh et al., 2017*; *Kawauchi et al., 2013*). However, except for the characterization of *Magnaporthe oryzae* MoSir2 in biotrophic growth in host rice plants (*Fernandez et al., 2014*), relatively little is known about the function of SIRTs in plant pathogens.

In this study, we show that the *F. oxysporum* FoSir5, an ortholog of the human lysine deacetylase, possesses decrotonylase activities in vitro and in vivo. In mitochondria, FoSir5 interacts with and removes the crotonyl group from the E2 component dihydrolipoyllysine acetyltransferase (FoDLAT) of the pyruvate dehydrogenase complex (PDC), the activity of which links glycolysis to the tricarboxylic acid (TCA) cycle. The specific decrotonylation of K148 of FoDLAT by FoSir5 leads to decreased PDC activity and acetyl-CoA generation. FoSir5 is also distributed in nuclei, where it directly regulates the enrichment of crotonylated H3K18 (H3K18cr) on genes involved in the TCA cycle and electron transport chain (ETC) pathways. The integrated regulation by FoSir5 in different organelles represses mitochondrial ATP biosynthesis, impeding fungal conidial germination. Importantly, we find that by decreasing the expression of *FoSir5*, *F. oxysporum* increases the ATP supply that supports the energy consumption demands of germination, emphasizing the importance of this regulatory process. These findings provide a clear example in which plant pathogenic fungi control conidial germination through an exquisite regulatory network that links metabolic activity to the developmental program of infection.

## Results

### FoSir5 has both mitochondrial and extra-mitochondrial decrotonylase activity in *vitro*

Members of the sirtuin family lysine deacetylases (KDACs) exhibit various subcellular localizations and are distributed in the nucleus, cytoplasm, and mitochondria (*North and Verdin, 2004*). However, none of the sirtuin KDACs in *F. oxysporum* have yet been characterized. The *F. oxysporum* genome contains seven genes predicted to encode a protein with the NAD$^+$-dependent deacetylase domain typical of the sirtuin KDACs, the same number found in the genomes of human and the fungus *N. crassa* (*Michishita et al., 2005*; *Smith et al., 2008*). We designated these genes *FoSir1* to *FoSir7* (*Figure 1A*).

The *FoSir5* (FOXG_05932) gene encodes a 298-amino acid protein that is predicted partition to the fungal mitochondrion (using WoLF PSORT). Similar to NST-6 (*Smith et al., 2008*), FoSir5 was identified as the closest homolog to human mitochondria-localized SIRT5 (*Figure 1B*). To confirm the subcellular localization of FoSir5, *FoSir5* cDNA was fused with green fluorescent protein (GFP) and transformed into *F. oxysporum*. As shown in *Figure 1C*, GFP-tagged FoSir5 partially colocalized with a mitochondrial fluorophore, and a detectable FoSir5 fraction was also present outside the organelle in the cytosol. Consistent with the confocal microscopy results, subcellular fractionation revealed that a significant fraction of FoSir5-GFP was present in the mitochondrion and cytosol (*Figure 1D*). The analysis also reveals a small fraction of FoSir5 in the nuclear fraction. Like *F. oxysporum* FoSir5, the human ortholog SIRT5 is also found both inside and outside the mitochondrion (*Rardin et al., 2013*).

Three human sirtuins, SIRT1-SIRT3, were recently suggested to remove crotonyl groups from histones in vitro (*Bao et al., 2014*). To investigate whether the FoSir5 protein possesses similar activity, we incubated bacterially expressed and purified recombinant FoSir5 with native calf thymus histones (CTH) in the presence of crotonyl-CoA. The pan anti-Kcr antibody specifically recognizing crotonylated lysine residues (*Liu et al., 2017*; *Tan et al., 2011*) was used for Western blotting to detect Kcr signal. Histone Kcr was detected in the untreated CTH samples, in agreement with earlier studies (*Sabari et al., 2015*), and the addition of FoSir5 resulted in a decrease in histone Kcr (*Figure 1E*), indicating that FoSir5 is able to remove crotonyl groups from histones in vitro.

To investigate whether *FoSir5* is involved in regulating *F. oxysporum* growth, we assessed the expression of the gene during different growth phases by quantitative real-time PCR (qRT-PCR). As shown in *Figure 1F*, the expression of *FoSir5* was high in the conidia, decreased dramatically during the germination process (4–12 hr), and then recovered in the mycelium at 24 hr. These differential expression patterns suggest that *FoSir5* might play a role in the conidial germination of *F. oxysporum*.

### FoSir5 modulates PDC activity by decrotonylating the E2 component of PDC

Since a large proportion of FoSir5 was localized in the mitochondria, we explored whether FoSir5 is involved in the decrotonylation of mitochondrial proteins in *F. oxysporum*. For this purpose, we utilized the FoSir5-GFP transformant and performed immunoprecipitation followed by liquid chromatography-tandem mass spectrometry (LC-MS/MS). Among the candidate-binding partners (*Supplementary file 1*), the E2 component of the PDC, FoDLAT, a putative mitochondrial protein (FOXG_11462), was selected for further analysis.

Fluorescence observation verified the expected mitochondrial localization of FoDLAT in *F. oxysporum* (*Figure 2—figure supplement 1A*). FoSir5-GFP and FoDLAT-Flag fusion constructs were co-introduced into *F. oxysporum* protoplasts, and positive transformants were selected. FoDLAT was detected in the proteins that eluted from anti-GFP beads using the anti-Flag antibody, suggesting that FoDLAT interacts with FoSir5 (*Figure 2A*). This interaction was also confirmed by in vitro pull-down assay, indicating a direct interaction between the two proteins (*Figure 2B*).

To test whether FoSir5 decrotonylates FoDLAT, we first generated ΔFoSir5 deletion mutants by replacing the coding region with the hygromycin resistance cassette (HPH) and FoSir5 overexpression strains fused with a C-terminal Flag tag driven by the strong constitutive promoter RP27. A total of three transformants from each group were obtained (*Figure 2—figure supplement 2*). The transformants from all groups had the same phenotypes, although only data for the mutant strain ΔFoSir5.3 (ΔFoSir5) and the overexpression strain FoSir5-Flag-1 (OE-1) are presented below. Then, we

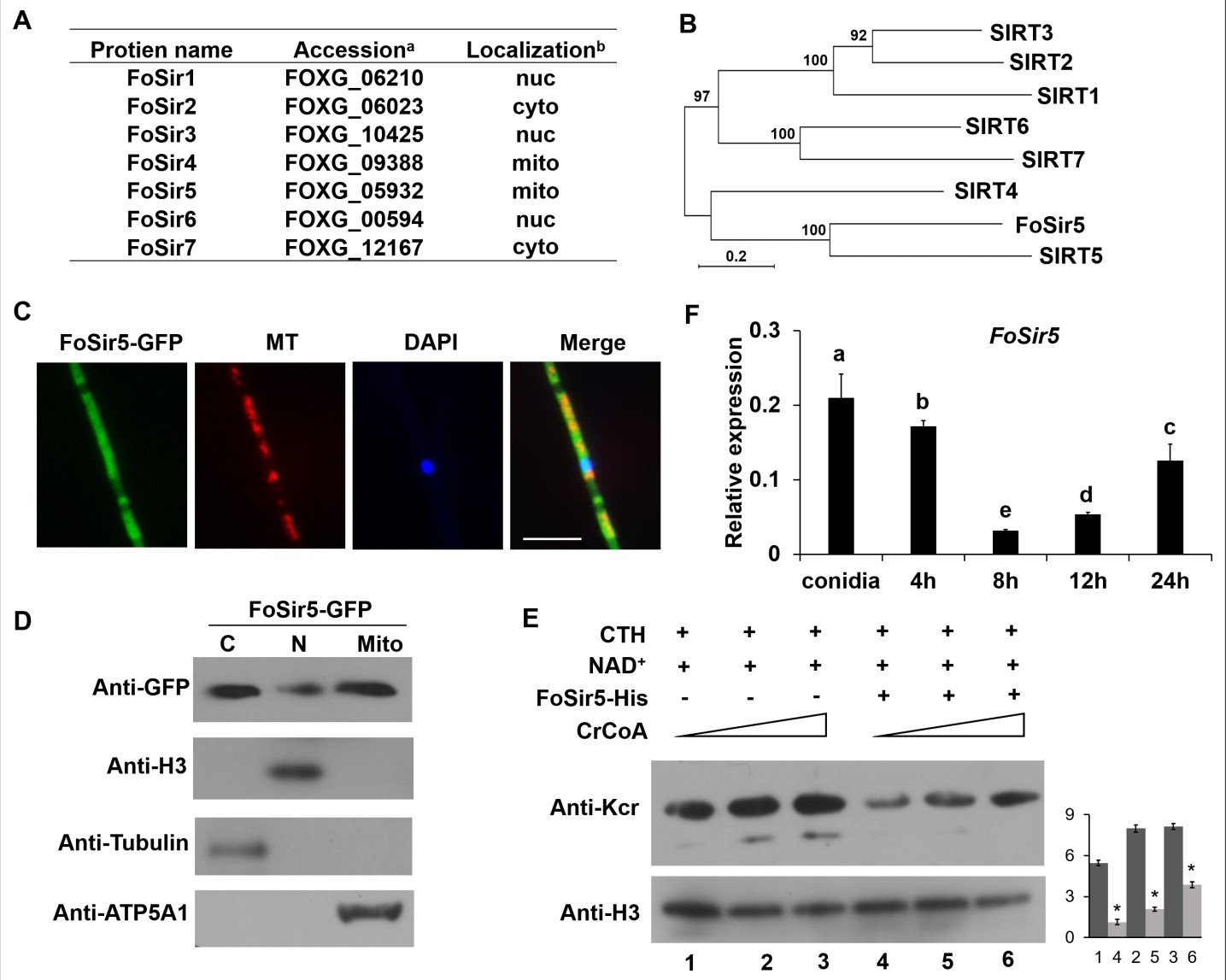

**Figure 1.** Cellular localization and activity of FoSir5 in *Fusarium oxysporum*. (**A**) Sirtuin proteins in *F. oxysporum* with predicted subcellular localizations. [a] Accession number of the full-length protein sequence available at Ensembl. [b]Localization of the *F. oxysporum* Sir2 protein determined by WoLF PSORT. (**B**) Phylogenetic tree relating FoSir5 to the orthologous human Sirtuin isoforms SIRT1 (NP_036370), SIRT2 (NP_085096), SIRT3 (NP_001357239), SIRT4 (NP_036372), SIRT5 (NP_001363737), SIRT6 (NP_057623), and SIRT7 (NP_057622). The tree is based on neighbor-joining analysis using MEGA-X. (**C**) Fluorescence microscopy analysis of FoSir5-GFP localization with MitoTracker Red (MT) and DAPI. Scale bars = 10 µm. (**D**) Subcellular fractionation of FoSir5-GFP transformants in *F. oxysporum*. Nuclear, cytoplasmic, and mitochondrial proteins were separately extracted and FoSir5-GFP were detected with anti-GFP antibody (Materials and methods). The fractionation controls were ATP5A1 (mitochondria), tubulin (cytosol), and histone H3 (nucleus). C, cytosol; N, nucleus; Mito, mitochondria. (**E**) In vitro Kcr assays with 50 µg of native calf thymus histone (CTH), 5 mM $NAD^+$, and 0.5 µg of FoSir5-His in the presence of 50, 100, or 200 µM crotonyl-CoA. Reaction materials were analyzed by Western blotting with anti-Kcr or anti-H3 antibody. Each scale bar represents the mean ± SD for triplicate experiments. * indicates a significant difference between different pairs of samples ($p < 0.05$). (**F**) Expression profile of *FoSir5* in conidia, mycelium, and during the germination process. The expression levels were normalized to that of the *F. oxysporum* elongation factor one alpha (EF-1α) gene. The presence of different letters above the mean values of three replicates indicates a significant difference between different samples ($p < 0.05$, ANOVA).

The online version of this article includes the following source data for figure 1:

**Source data 1.** Cellular localization and activity of FoSir5 in *Fusarium oxysporum*.

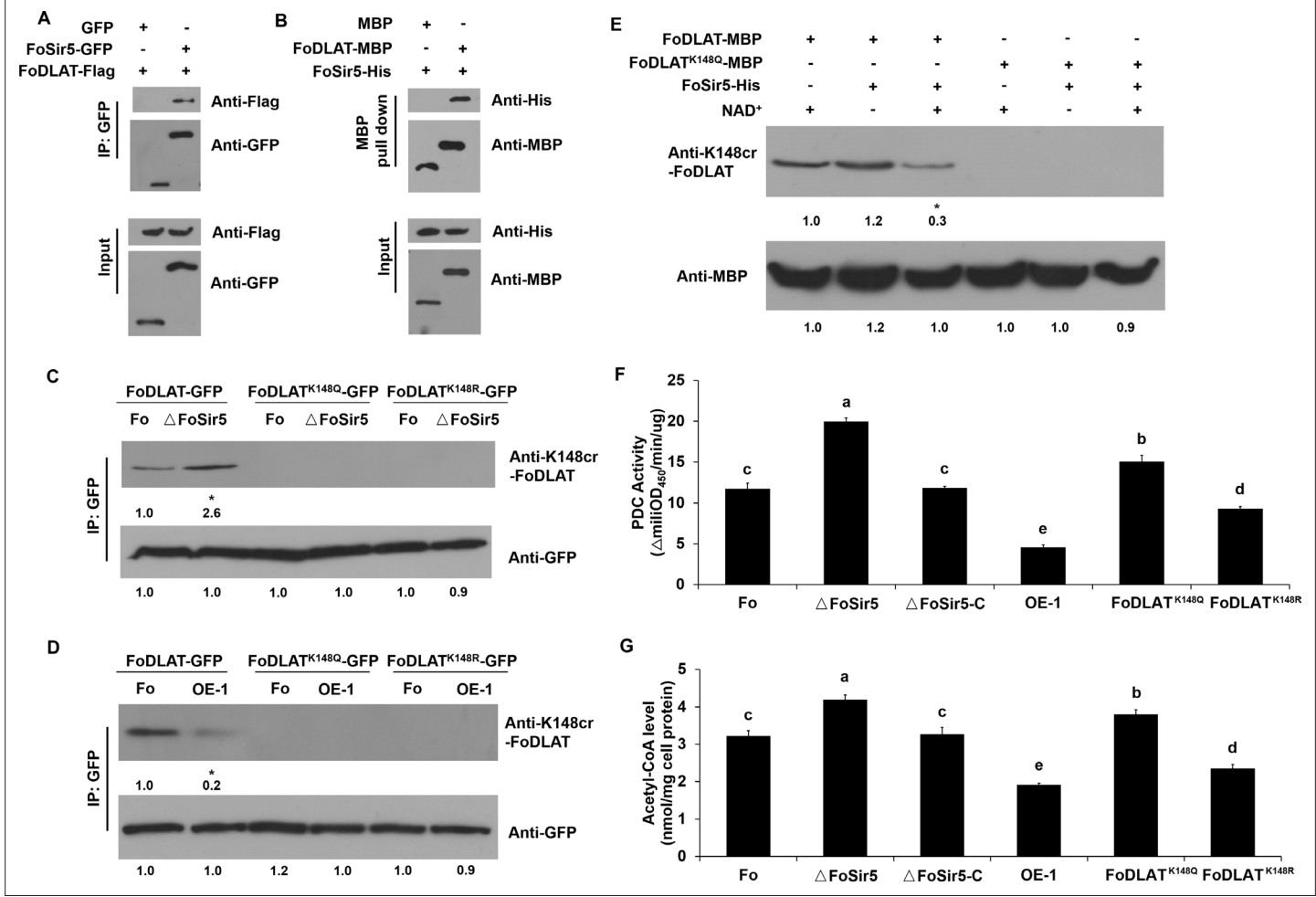

**Figure 2.** FoSir5 decrotonylates FoDLAT, the E2 component of *Fusarium oxysporum* pyruvate decarboxylase complex, with regulatory consequence. (**A**) Co-IP assays reveal physical interaction of FoSir5-GFP and FoDLAT-Flag. Western blot analysis of cell extracts from transformants co-expressing FoDLAT-Flag with GFP or FoSir5-GFP and elution from anti-GFP agarose. The fusion proteins were detected with anti-Flag or anti-GFP antibody. (**B**) In vitro pull-down assays to detect FoSir5-His with MBP or the FoDLAT-MBP fusion protein. FoDLAT-MBP was used as bait to pull down the FoSir5-His protein from the induced cell extracts. The MBP protein was assayed as a negative control. Input and bound forms of the pull-down fractions were detected with anti-His or anti-MBP antibody. (**C–D**) The K148 crotonylation (anti-K148cr-FoDLAT, top panel) and amount (anti-GFP, bottom panel) of FoDLAT-GFP and its mutant isoforms in the ΔFoSir5 (**C**) and OE-1 strain (**D**). Proteins were immunoprecipitated with anti-GFP antibody agarose beads and analyzed by anti-K148cr-FoDLAT or anti-GFP antibody. Representative gels are shown from experiments carried out at least twice. Numbers below the blots represent the relative abundance of K148-crotonylated FoDLAT. Anti-GFP immunoblotting was used to show equal loading. (**E**) FoSir5 directly decrotonylates FoDLAT in vitro. Purified FoDLAT protein or its K148Q isoform (50 ng) were incubated with or without 50 ng of purified FoSir5 in the absence or presence of 5 mM NAD$^+$ and then analyzed by immunoblotting using anti-K148cr-FoDLAT or anti-His antibody. Each gel shown is representative of two experiments. Numbers below the blots represent the relative abundance of K148-crotonylated FoDLAT. Anti-MBP immunoblotting was used to show equal loading. (**F–G**) FoSir5 and K148 mutant FoDLAT affected pyruvate dehydrogenase complex (PDC) activity (**F**) and acetyl-CoA production (**G**) in *F. oxysporum*. PDC activity and acetyl-CoA production were determined in germinating conidia at 8 hr. The presence of different letters above the mean values of three replicates indicates a significant difference between different strains (p < 0.05, ANOVA).

The online version of this article includes the following source data and figure supplement(s) for figure 2:

**Source data 1.** FoSir5 decrotonylates FoDLAT, the E2 component of *Fusarium oxysporum* pyruvate decarboxylase complex, with regulatory consequence.

**Figure supplement 1.** Interpretation of the subcellular location, Kcr site, and protein structure of FoDLAT.

**Figure supplement 2.** Generation of targeted *FoSir5* gene deletion mutants and overexpression transformants.

**Figure supplement 2—source data 1.** Generation of targeted *FoSir5* gene deletion mutants and overexpression transformants.

**Figure supplement 3.** Detection of crotonylation, succinylation, malonylation, and glutarylation on FoDLAT protein in ΔFoSir5 compared with Fo.

**Figure supplement 3—source data 1.** Detection of crotonylation, succinylation, malonylation, and glutarylation on FoDLAT protein in ΔFoSir5 compared with Fo.

transformed and expressed FoDLAT-GFP in the Fo, ΔFoSir5, and OE-1 strains. The crotonylation levels of immunoprecipitated FoDLAT were then tested. Compared with that in the Fo strain, we found markedly increased crotonylation of FoDLAT in ΔFoSir5 (*Figure 2—figure supplement 3*). These data indicated that FoSir5 is responsible for the decrotonylation of FoDLAT. Although SIRT5 was reported to possess robust demalonylase, desuccinylase, and deglutarylase activities in mammals (*Hirschey and Zhao, 2015*; *Park et al., 2013*), our results showed that FoSir5 had no detectable impact on succinylation, malonylation, or glutarylation of the FoDLAT protein in *F. oxysporum* (*Figure 2—figure supplement 3*).

To determine the crotonylation sites of FoDLAT, we purified the FoDLAT-GFP fusion protein from *F. oxysporum* and identified by mass spectrometry lysine 148 as a site of modification (*Figure 2—figure supplement 1B*). To further confirm whether K148 is crotonylated in vivo, we mutated lysine 148 to arginine (R) or glutamine (Q), respectively, and developed a specific antibody against crotonylated K148 of FoDLAT (anti-K148cr-FoDLAT) to examine their crotonylation. Note that arginine and glutamine mimic non-acylated and acylated lysine, respectively, with respect to charge on the residues (*Sun et al., 2020*; *Yu et al., 2020*). The Western analyses detected signals with only the wild-type (WT) FoDLAT but not the Q and R mutants, indicating that K148 is indeed crotonylated in vivo. We also found that inactivation and overexpression of FoSir5 led to significantly increased and decreased K148 crotonylation of WT FoDLAT, respectively (*Figure 2C and D*). To further verify that FoSir5 plays a role in the decrotonylation of FoDLAT, an in vitro decrotonylation assay was performed using recombinant proteins purified from *Escherichia coli*. The results showed that K148 crotonylation has also occurred on FoDLAT in *E. coli* and addition of FoSir5 in the presence of NAD$^+$ reduced K148 crotonylation of WT FoDLAT but not the Q form (*Figure 2E*). These data demonstrate that FoSir5 is responsible for K148 decrotonylation of FoDLAT.

To explore how this crotonylation site might affect FoDLAT function, we first generated a homology model of the fungal enzyme based on the crystal structure of the homologous human E2. In the PDC assembly, E2 is the dihydrolipoyl acetyltransferase component, and comprises a biotin-lipoyl domain, an interaction domain that binds the dihydrolipoyl dehydrogenase (E3) component, and the catalytic domain. The crotonylation site maps to an intradomain linker that is predicted to be flexible (*Figure 2—figure supplement 1C* and D). The flexibility enables the biotin-lipoyl domain to shuttle substrates between the E1 and E2 catalytic sites and then to the E3 site for an oxidation step. Given that DLAT is likely an essential component of PDC function (no DLAT deletion mutants could be obtained after numerous attempts in this research), we examined the impact of changed FoSir5 levels on the endogenous cellular activity of the PDC. PDC activity was elevated in the ΔFoSir5 strain and reduced in the OE-1 strain compared with the Fo and the complemented strain ΔFoSir5-C (*Figure 2F*). To further determine whether the crotonylation site of FoDLAT plays a role in PDC function, we detected PDC activity in the K148 mutant strains. As shown in *Figure 2F*, the K148Q and K148R mutant strains demonstrated increased and decreased PDC activity, respectively. Furthermore, the levels of acetyl-CoA, a direct product of E2 catalytic activity, followed a pattern similar to that of PDC activity among the different strains (*Figure 2G*). Collectively, these data establish a specific and prominent role of FoSir5 in FoDLAT decrotonylation and PDC enzymatic inactivation.

## FoSir5 directly regulates the expression of genes related to aerobic respiration through H3K18 decrotonylation

Subcellular fractionation showed a small portion of FoSir5 in the nuclei, and FoSir5 could remove crotonyl groups from histones in vitro (*Figure 1C–E*). This finding lead us to explore whether FoSir5 regulates histone Kcr in cells. As shown in *Figure 3A*, FoSir5 inactivation caused the accumulation of H3K18cr, but has little effect on H3K18ac or H3K9cr. Therefore, we performed RNA-seq analysis to detect transcripts that might be regulated by FoSir5 in *F. oxysporum*. Three biological replicates consisting of mRNA isolated from the Fo and ΔFoSir5 mutant strains were assessed, identifying 1566 up- and 856 downregulated (fold change >2, p < 0.05) genes in the ΔFoSir5 compared with the Fo strain (*Supplementary file 2*). Given the function of histone decrotonylases (HDCRs) in gene repression and H3K18cr in gene activation (*Sabari et al., 2015*), we hypothesized that the target genes of FoSir5 were likely among the upregulated genes. Gene Ontology (GO) functional annotation and KEGG pathway analysis of those upregulated genes revealed significant enrichment of their products in the TCA cycle, ETC, and ATP synthesis (*Figure 3—figure supplement 1*). We selected for experimental

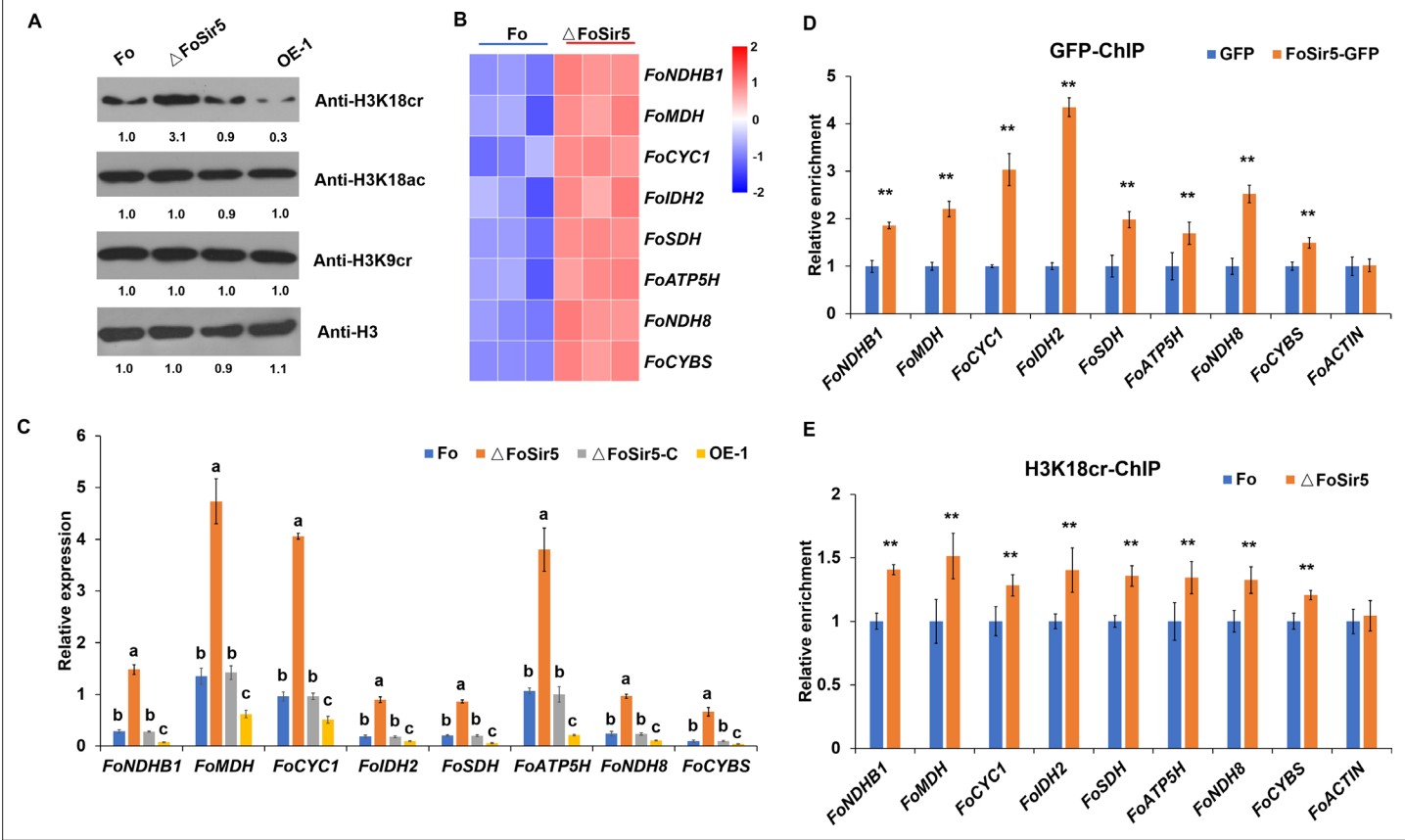

**Figure 3.** The downregulation of H3K18 crotonylation by FoSir5 and transcriptional repression of aerobic respiration-related genes. (**A**) Western blot analysis showed the effect of FoSir5 on histone H3K18 crotonylation and acetylation, and histone H3K9 crotonylation using the indicated antibodies. Numbers below the blots represent the relative abundance of different modifications. Anti-H3 immunoblotting was used to show equal loading. (**B**) RNA-seq analysis of eight upregulated genes involved in aerobic respiration including *NDHB1* (NADH-quinone oxidoreductase chain B 1), *MDH* (malate dehydrogenase), *CYC1* (cytochrome C1), *IDH2* (isocitrate dehydrogenase subunit 2), *SDH* (succinate dehydrogenase), *ATP5H* (ATP synthase D chain), *NDH8* (NADH dehydrogenase iron-sulfur protein 8), and *CYBS* (succinate dehydrogenase cytochrome b small subunit). Differential expression in three biological replicates illustrated using a heat map with colored squares indicating the range of expression referred to as the FPKM value. (**C**) qRT-PCR validation of aerobic respiration-related genes in the indicated strains. The letters above the mean values of three replicates indicate significant differences between different strains (p < 0.05, ANOVA). (**D–E**) Relative enrichment of the immunoprecipitated promoter regions in aerobic respiration-related genes determined using anti-GFP antibody in the FoSir5-GFP strain and Fo strain containing GFP alone (**D**) or using anti-H3K18cr antibody in the Fo and ΔFoSir5 mutant strains (**E**). The fold enrichment was normalized to the input and internal control gene (*β-tubulin*). Data are the means ± SDs (n = 3); **p < 0.05 by unpaired two-tailed t-test.

The online version of this article includes the following source data and figure supplement(s) for figure 3:

**Source data 1.** The downregulation of H3K18 crotonylation by FoSir5 and transcriptional repression of aerobic respiration-related genes.

**Figure supplement 1.** Distribution of functional classification of Gene Ontology (GO) (**A**) and KEGG pathway (**B**) of the upregulated genes in ΔFoSir5 compared with Fo.

validation eight upregulated genes related to types of energy metabolism, namely *NDHB1* (NADH-quinone oxidoreductase chain B 1), *MDH* (malate dehydrogenase), *CYC1* (cytochrome C1), *IDH2* (isocitrate dehydrogenase subunit 2), *SDH* (succinate dehydrogenase), *ATP5H* (ATP synthase D chain), *NDH8* (NADH dehydrogenase iron-sulfur protein 8), and *CYBS* (succinate dehydrogenase cytochrome b small subunit) (*Figure 3B*). qRT-PCR analysis indicated that all of the eight genes were indeed upregulated in the ΔFoSir5 strain and downregulated in the OE-1 strain compared with the Fo and ΔFoSir5-C strains (*Figure 3C*).

To determine whether FoSir5 directly regulates these eight genes, a chromatin immunoprecipitation (ChIP) qPCR assay was performed using a GFP antibody. Primers in promoter regions near putative transcription start sites (TSSs) were designed to evaluate the enrichment of FoSir5-GFP in the eight energy metabolism-related genes. The results showed that these regions were highly enriched

by FoSir5 in the FoSir5-GFP strain compared with the Fo strain (*Figure 3D*). To test whether these promoter regions are also H3K18 crotonylation locations in genomic DNA, we further performed ChIP using an anti-H3K18cr antibody, followed by qPCR. As shown in *Figure 3E*, these regions were also enriched by H3K18cr in the ΔFoSir5 compared with the Fo strain. *FoACTIN* was used as a negative control which was not enriched by anti-GFP or anti-H3K18cr (*Figure 3D and E*). Overall, these observations demonstrated that FoSir5 and H3K18cr were enriched in the promoter regions of eight energy-related genes, indicating that FoSir5 and H3K18cr participate in the transcriptional regulation of metabolic energy-generating systems in *F. oxysporum*.

## FoSir5 represses ATP synthesis in germinating *F. oxysporum*

Most intracellular ATP comes from the oxidation of glucose-derived pyruvate by the TCA cycle and oxidation of NADH in mitochondria via the ETC. As genes involved in mitochondrial ATP synthesis were directly regulated by FoSir5, we speculated that change of FoSir5 level will result in altered ATP content. As shown in *Figure 4A–E*, dramatic decrease of FoSir5 during germination led to reduced decrotonylase activity of this enzyme, and as a result, the K148 crotonylation of FoDLAT, PDC activity, acetyl-CoA generation, H3K18cr level, and expression of the eight energy-related genes were elevated. Meanwhile, declined enrichment of FoSir5 in promoter regions of these genes was observed (*Figure 4F*). Ultimately, the ATP content was elevated during the germinating process (*Figure 4G*). Consistent with these observations, inactivation of FoSir5 increased the level of ATP by ~70%, whereas overexpression of this enzyme significantly decreased ATP content in germinating conidia at 8 hr post incubation (*Figure 4H*). Moreover, the FoSir5 mutant and overexpression strains exhibited a continuous high and low level of ATP during the whole germinating process, respectively (*Figure 4—figure supplement 1*), further confirming the relationship between FoSir5 and ATP.

## FoSir5 affects conidial germination of *F. oxysporum* by modulating ATP generation

Previous studies demonstrated that ATP plays a significant role in energizing cellular developmental processes (*Wang et al., 2013*). As conidial germination is of high energy consumption, it is reasonable to envision that elevated ATP level benefits this process. Therefore, we determined germination rates of conidia treated with exogenous ATP at different concentrations from 0 to 50 µM. Not unexpectedly, supply with at least 10 µM ATP increased germination rate of *F. oxysporum* conidia by about 50% (*Figure 5A*).

Modulation of ATP levels of *F. oxysporum* to support germination is likely to involve an extensive regulatory network, and the sharp decline of FoSir5 during germination (*Figure 1F*) might be expected to be important for this process. To examine this point in more detail, we tested conidial germination of Fo, ΔFoSir5, and ΔFoSir5-C mutant strains. We found that while inactivation of FoSir5 elevated conidial germination, reintroduction of FoSir5 recovered the phenotype of the WT strain. Conversely, overexpression of FoSir5 led to obviously decreased germination (*Figure 5B and C*). Addition of exogenous ATP completely rescued the impeded germination of the OE-1 strain (*Figure 5D*), confirming an important role of ATP in conidial germination. In support of these observations, either K148Q mutation of FoDLAT or overexpression of key genes of the TCA cycle and ATP metabolism including *MDH*, *ATP5H*, and *CYC1* (*Figure 5—figure supplement 1*) led to elevated ATP and conidial germination, whereas the K148R mutant strain demonstrated decreased ATP level and germination (*Figure 5E and F*). Taken together, all of these data strongly suggest a crucial role for FoSir5 in conidial germination through modulating ATP generation.

Based on these results and those presented above, we propose a simple model to explain how FoSir5 modulates conidial germination of *F. oxysporum* (*Figure 6*). In mitochondria, FoSir5 binds and decrotonylates FoDLAT at K148, and this modification inhibits the enzymatic activity of PDC leading to reduced production of acetyl-CoA. At the same time, FoSir5-catalyzed H3K18cr decrotonylation in the nucleus transcriptionally represses the expression of genes participating in the TCA cycle and ETC pathways with acetyl-CoA being the initial substrate. Consequently, the coordinated regulation by FoSir5 in different organelles results in the repression of mitochondrial ATP synthesis. During conidial germination, by decreasing FoSir5 level, *F. oxysporum* eliminates inhibition of ATP metabolism essential for this process.

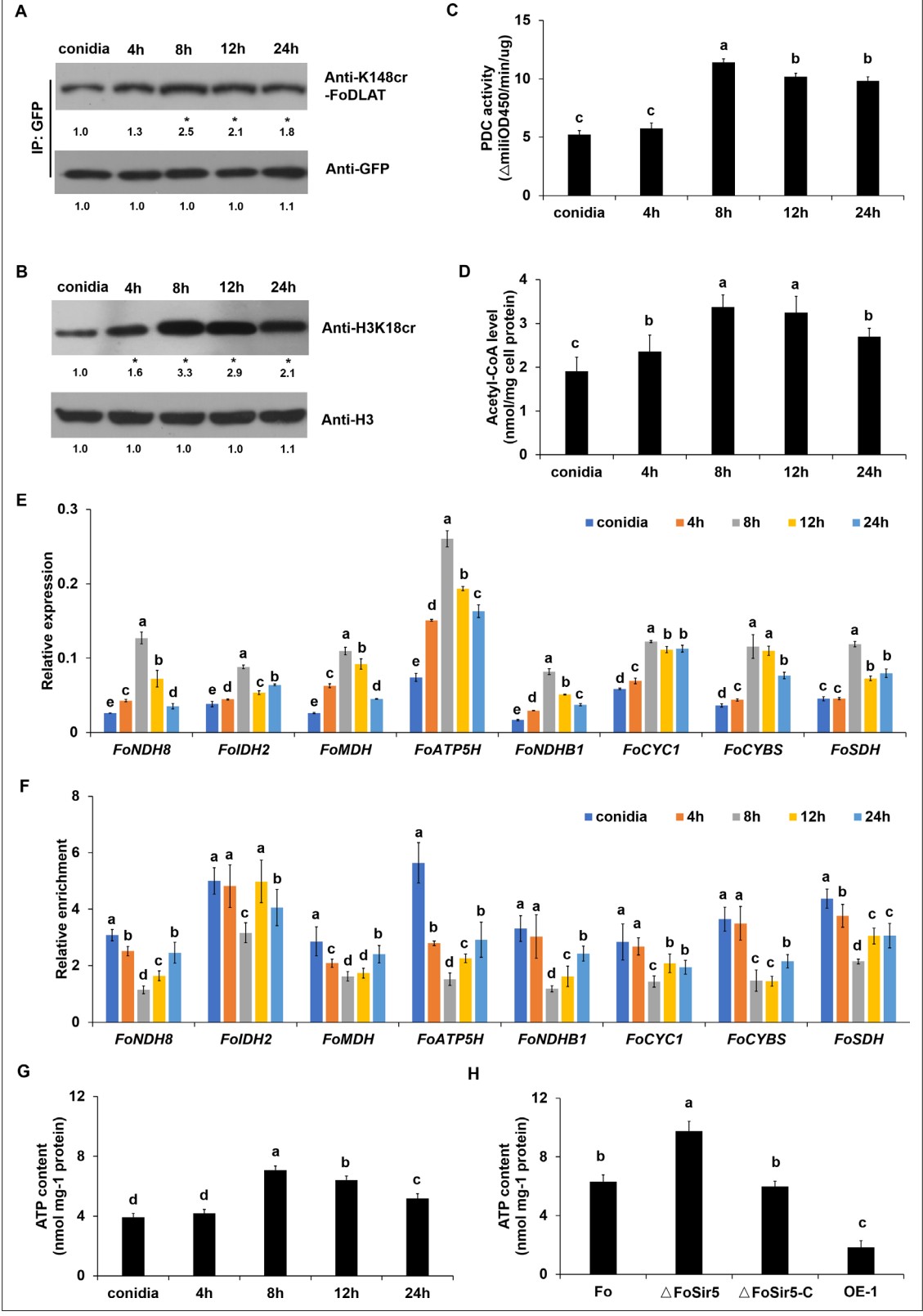

**Figure 4.** FoSir5 affects ATP production during gemination in *Fusarium oxysporum*. (**A–B**) Western blot analysis showed the dynamic changes of FoDLAT K148 (**A**) and histone H3K18 (**B**) crotonylation during germination using the indicated antibodies. Numbers below the blots represent the relative abundance of FoDLAT-K148cr or H3K18cr. Anti-GFP or anti-H3 immunoblotting was used to show equal loading, respectively. (**C–D**) Pyruvate dehydrogenase complex (PDC) activity (**C**) and acetyl-CoA production (**D**) in *F. oxysporum* during germination were determined. (**E**) Expression profile

*Figure 4 continued on next page*

*Figure 4 continued*

of the aerobic respiration-related genes during the germination process. (**F**) Relative enrichment of the immunoprecipitated promoter regions in aerobic respiration-related genes during germination determined using anti-GFP antibody in the FoSir5-GFP strain driven by the native promoter. The fold enrichment was normalized to the input and internal control gene ($\beta$-*tubulin*). (**G**) ATP content of *F. oxysporum* during germination. (**H**) Effect of FoSir5 on the ATP content of the indicated strains, as determined in germinating conidia at 8 hr post incubation (h.p.r.). The presence of different letters (**A–H**) above the mean values of three replicates indicates a significant difference between different samples ($p < 0.05$, ANOVA).

The online version of this article includes the following source data and figure supplement(s) for figure 4:

**Source data 1.** FoSir5 affects ATP production during gemination in *Fusarium oxysporum*.

**Figure supplement 1.** The ATP content of ΔFoSir5 mutant (**A**) and OE-1 strain (**B**) during germinating process.

**Figure supplement 2.** Impact of FoSir5 on the virulence of *Fusarium oxysporum*.

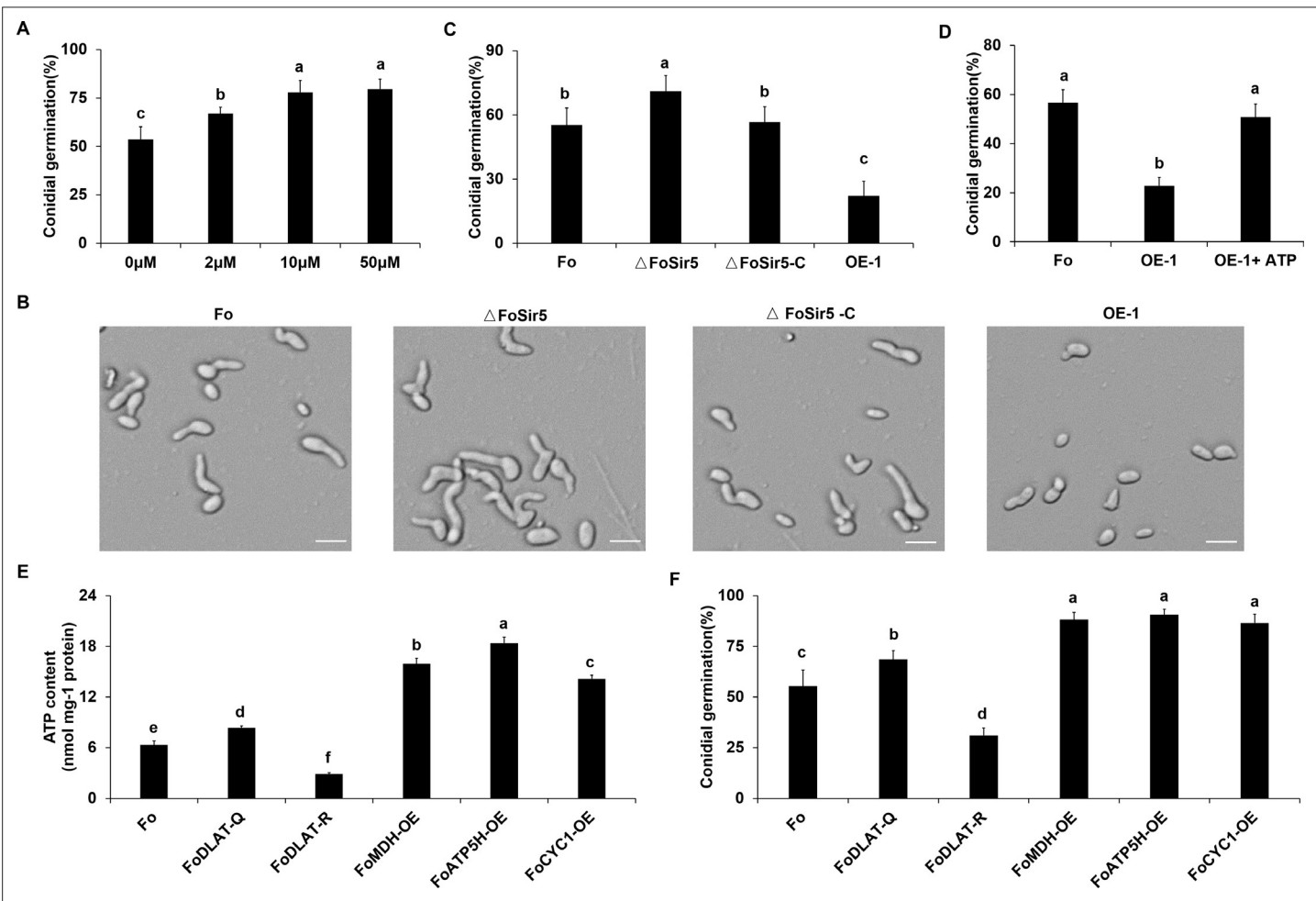

**Figure 5.** FoSir5 modulates conidial germination through affecting ATP synthesis. (**A**) Quantification of the conidial germination of *Fusarium oxysporum* in PDB supplied with different concentrations of exogenous ATP at 8 h.p.r. (**B**) Conidial germination of the indicated strains in PDB on glass slides at 8 h.p.r. Representative images from three or more independent experiments, all of which had similar results. Scale bars = 30 µm. (**C**) Quantification of the conidial germination of the indicated strains in PDB on glass slides at 8 h.p.r. (**D**) Quantification of the conidial germination of the OE-1 strain with or without treatment with exogenous ATP at 8 h.p.r. (**E–F**) Effect of FoDLAT -K148Q/R mutations or overexpression of key genes of aerobic respiration on ATP production (**E**) and conidial germination (**F**) in *F. oxysporum*. The ATP content and germinating rate were determined at 8 h.p.r. The letters (**A–F**) above the mean values of three replicates indicate significant differences between different strains ($p < 0.05$, ANOVA).

The online version of this article includes the following figure supplement(s) for figure 5:

**Figure supplement 1.** Real-time PCR (RT-PCR) analysis of *FoMDH-*, *FoATP5H-*, and *FoCYC1*-overexpressing transformants.

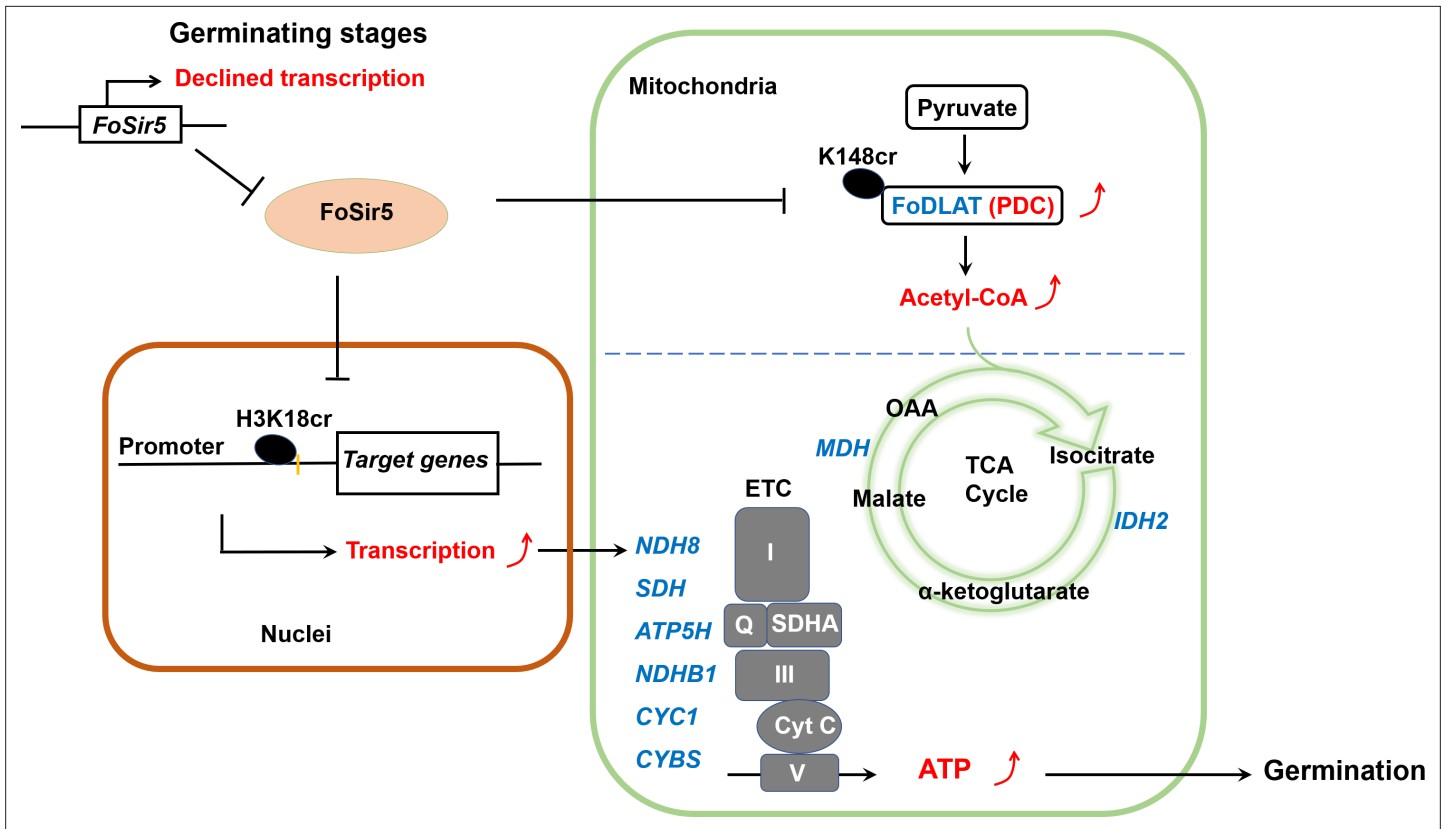

**Figure 6.** A model for FoSir5 functioning as a decrotonylase in different organelles to regulate conidial germination. During the germination process, the expression of *FoSir5* decreases, leading to relief of the inhibitory effect of FoSir5 on pyruvate dehydrogenase complex (PDC) activity through the decrotonylation of FoDLAT-K148cr and transcription of aerobic respiration-related genes by the reversal of H3K18cr. Thus, mitochondrial ATP biosynthesis is enhanced, promoting conidial germination in *Fusarium oxysporum*.

## FoSir5 is required for full virulence of *F. oxysporum* on tomato

To determine whether FoSir5-mediated ATP metabolism affects pathogenicity of *F. oxysporum*, infection assays were performed by dipping the roots of 2-week-old tomato seedlings in conidial suspension of the Fo, ΔFoSir5, ΔFoSir5-C, and OE-1 strains. In our repeated experiments, compared with Fo and ΔFoSir5-C, the ΔFoSir5 strain exhibited higher infection ability, whereas overexpression of *FoSir5* reduced disease development by ~40% (*Figure 4—figure supplement 2A* and B). To determine whether altered level of FoSir5 affected in planta fungal growth, we quantified fungal biomass in roots by analyzing the expression level of *F. oxysporum FoEF-1α* as an indicator. Consistent with the results of infection assays, the level of *FoEF-1α* in ΔFoSir5 inoculated plants increased almost 2-fold, and the amount of fungal transcript was reduced by ~70% in OE-1 infected roots (*Figure 4—figure supplement 2C*). These results indicate that FoSir5 exerts a negative effect on virulence of *F. oxysporum*, and the increased and decreased pathogenicity of ΔFoSir5 and OE-1, respectively, is likely due to increased and decreased germination rates.

## Discussion

Lysine crotonylation, a newly discovered PTM reversibly controlled by lysine crotonytransferases and decrotonylases, is involved in numerous cellular processes, including chromatin remodelling, metabolism, protein folding, and the cell cycle (*Wan et al., 2019*; *Xu et al., 2017*). Although a growing number of crotonylated proteins have been identified in multiple organisms (*Kwon et al., 2018*; *Liu et al., 2018*; *Sun et al., 2017*; *Sun et al., 2019*; *Zhang et al., 2020a*), the enzymes responsible for lysine crotonylation and their physiological role remain poorly defined, especially for the decrotonylation of non-histone proteins. The information presented here indicates that a sirtuin family protein, FoSir5, functions as a lysine decrotonylase to modulate conidial germination in *F. oxysporum*. As such,

these findings greatly expand our understanding of lysine decrotonylases and open up new possibilities for further investigations of the regulatory role of these enzymes.

The studies described here provide evidence that FoSir5 can modulate ATP synthesis through lysine decrotonylation in different organelles and thus conidial germination of *F. oxysporum*. Specifically, we found that (1) FoSir5 is distributed in different cellular compartments and possesses lysine decrotonylase activity; (2) in mitochondrion, FoSir5 removes a crotonyl group from the K148 residue of FoDLAT, and therefore inhibits PDC activity and acetyl-CoA production; (3) in the nucleus, FoSir5 represses the expression of genes associated with aerobic respiration by decrotonylating H3K18cr; (4) the coordinated regulation by FoSir5 in the mitochondrion and nucleus impacts on ATP synthesis; (5) ATP content is positively associated with conidial germination; and (6) *F. oxysporum* downregulates *FoSir5* level during germination to elevate ATP level required for this process. These findings indicate that fungal pathogens employ an elaborate mechanism to carefully control energy metabolism. The mechanism by which FoSir5 is downregulated during germination is a topic to be explored in future studies, but it is clear that this complex regulatory system serves as a salient example of how eukaryotes can control their development through regulating the action of a lysine decrotonylase.

Sirtuins are class III KDACs that require NAD for their deacylation activities. Seven sirtuin isoforms (SIRT1-SIRT7) are expressed in mammalian cells. These isoforms display widespread subcellular distributions, as SIRT1, SIRT6, and SIRT7 are nuclear, SIRT2 is predominantly cytoplasmic, and SIRT3-5 are mitochondrial (*Gertz and Steegborn, 2016*; *Michishita et al., 2005*). Recent studies have shown both the mitochondrial and extra-mitochondrial localization of SIRT5 (*Park et al., 2013*), while SIRT1 and SIRT2 can accumulate in the cytosol and nucleus, respectively, under specific circumstances (*Byles et al., 2010*; *Vaquero et al., 2006*). However, the synergistic action of sirtuins among different organelles is poorly characterized. Our findings that SIRT5 simultaneously act on histones in chromatin and enzymes in the mitochondria to modulate ATP generation provide a clear example of coordinated functions of one sirtuin protein in different cellular compartments. With the identification of more lysine deacylases in future research, it is likely that the findings reported here are only the beginning of what will be a widespread phenomenon in eukaryotes.

By converting pyruvate to acetyl-CoA, PDC is an important gatekeeper that links glycolysis to the TCA cycle and oxidative phosphorylation. Therefore, controlling the activity of this enzyme complex impacts on metabolic flux and the efficiency of ATP generation. In mammalian cells, pyruvate dehydrogenase phosphatases dephosphorylate the E1α subunit and activate the PDC, while SIRT5-mediated desuccinylation of PDC subunits, including mainly E1α, E1β, and E3, suppresses PDC activity (*Park et al., 2013*). The data described here provide evidence that FoSir5 decrotonylates the E2 subunit of the PDC at K148 and thus inhibits PDC activity in *F. oxysporum*. All these findings indicate that cells employ a variety of approaches to tightly regulate the activity of PDC. It will be of considerable interest to examine the coordinated effect exerted by different modifications in future studies.

Conidia are reproductive structures important for both dispersal and survival within harsh environments. In this study, we found that the expression level of *FoSir5* was higher in conidia than in other growth stages (*Figure 1F*), indicating that inhibition of mitochondrial ATP biosynthesis by FoSir5 may be helpful for maintaining low energy expenditure of conidia under unsuitable conditions. Thereafter, during the germination process needing high energy consumption, the expression of *FoSir5* sharply declined, resulting in enough ATP production to support breaking dormancy and the formation of a germ tube. Then, *FoSir5* levels were restored to a higher level in the mycelium, likely to control energy metabolism properly. Given the fact that conidial germination is crucial for infection and there is limited information on the regulation of this process (*Deng et al., 2015*; *Leroch et al., 2013*; *Sharma et al., 2016*), our findings that deletion of FoSir5 resulted in enhanced pathogenicity provide candidate target proteins for exploring new effective fungicides against *F. oxysporum* and other plant pathogens.

## Materials and methods
### Fungal strains and culture conditions
*F. oxysporum* f. sp. *lycopersici* strain 4287 (Fo) was used in all experiments. The fungus was stored at –80°C as microconidial suspension with 30% glycerol. It was grown on potato dextrose agar (PDA) at 25°C for 7 days in the dark to generate conidia. Spores were harvested using sterilized H$_2$O and

filtrated through four layers of sterile lens paper. Cultures were inoculated at a concentration of $1 \times 10^7$ conidia/ml in YPD medium (2% peptone, 1% yeast extract, and 2% glucose) at 25°C with shaking at 150 rpm. For conidial germination assay, fresh conidia of strains were harvested and adjusted to the concentration of $2.5 \times 10^5$ conidia/ml in PDB (liquid PDA). Twenty µl of the conidial suspension were dropped onto coverslips and incubated in a moist chamber with a temperature of 25°C. At least three independent experiments with triple replicates per experiment were performed.

## Target gene deletion, complementation, and overexpression

The *FoSir5* gene deletion mutant was generated using the standard one-step gene replacement strategy (*Figure 2—figure supplement 2A*). First, two fragments with 0.7 kb of sequences flanking the targeted gene were PCR-amplified with primer pairs UP-F/R and Down-F/R, respectively. Thereafter, the two flanking sequences were linked with an HPH by overlap PCR. The amplified fragment using primer pairs K-F/R was then purified and introduced into Fo protoplasts (*Gronover et al., 2001*; *Jiang et al., 2011*). Deletion mutants were identified by PCR with primer pairs IN-F/R and OUT-F/R. For complementation, a fragment encompassing the entire *FoSir5* gene coding region and its native promoter region was amplified by PCR with primers FoSir5-C-F/R and inserted into pYF11 (G418 resistance) vector by the yeast gap repair approach (*Bruno et al., 2004*; *Tang et al., 2020*). Then, the construct was used for protoplast transformation of the ΔFoSir5 mutant.

For site-directed mutagenesis of *FoDLAT*, we first tried to delete the *FoDLAT* gene, however, despite numerous attempts (over 200 transformants), we failed to obtain knockout mutants, indicating that FoDLAT disruption was lethal in Fo. Alternatively, *FoDLAT^K148Q* or *FoDLAT^K148R* genes with native promoter region was generated by fusion PCR using primer FoDLAT-C-F/R and cloned into pYF11 plasmid to form GFP fusion constructs. Then the constructs were transformed into protoplast of Fo. After verification by PCR and sequencing, deletion of *FoDLAT* was performed as described above to generate strains FoDLAT^K148Q and FoDLAT^K148R, respectively.

For construction of the RP27:FoSir5/FoMDH/FoATP5H/FoCYC1:GFP vectors, we amplified fragments by PCR with primer pairs GFP-F/R of each gene, respectively. The fragments were then inserted into the pYF11 vector (*Qi et al., 2016*). For construction of the RP27:FoSir5/FoDLAT:Flag vectors, fragments amplified with primers FoSir5-Flag-F/R or FoDLAT-Flag-F/R were inserted into pHZ126 vector (hygromycin resistance). The constructs were then used for protoplast transformation of Fo or other strains. The primers used in this study were listed in *Supplementary file 3*.

## Epifluorescence microscopy

*F. oxysporum* cells expressing FoSir5-GFP or FoDLAT-GFP fusion proteins were incubated on PDA plates at 25°C for 3 days. The mycelia of the tested strains were then collected and preincubated for 15 min with 200 nM MitoTracker Red CMRos (M7512, Invitrogen). After washing with phosphate-buffered saline (PBS), pH 7.4, the mycelia were stained with 1 µg/ml DAPI (D9542, Sigma) at room temperature in darkness for 5 min, followed by washing with PBS three times. Fluorescence microscope was performed using microscope of EVOS M5000 (Invitrogen).

## Subcellular fractionation analysis

The nuclear and cytosolic proteins were extracted using Nuclear Protein Extraction Kit (R0050, Solarbio) and mitochondrial proteins were extracted by Mitochondrial Extraction Kit (SM0020, Solarbio), according to the manufacturer's instructions. The obtained proteins were separated by SDS–PAGE and immunoblotted using anti-GFP (ab290, Abcam), anti-H3 (ab1791, Abcam), anti-Tubulin (PTM-1011, PTM Biolabs), and anti-ATP5A1 (459240, Thermo Fisher).

## In vitro HDCR assays

pET28 construct containing His-fused FoSir5 was expressed in BL21 *E. coli*. Protein expression was induced by adding isopropyl β-D-1-thiogalactopyranoside (IPTG) to a final concentration of 0.2 mM when OD600 reached 0.6, and the culture was further grown at 16°C overnight. Cells were harvested and resuspended in lysis buffer A (50 mM Tris-HCl, pH 7.5, 300 mM NaCl, 1 mM PMSF, and Roche EDTA free protease inhibitor). Following sonication and centrifugation, the supernatant was loaded onto a nickel column pre-equilibrated with lysis buffer. The column was washed with five column volumes of wash buffer (lysis buffer with 20 mM imidazole) and the bound proteins were then eluted

with elution buffer (lysis buffer with 200 mM imidazole). After purification, proteins were dialyzed at 4°C overnight. In vitro enzymatic reactions were performed as described previously (*Liu et al., 2017*). In brief, 50 µg native CTH (A002544, Sangon Biotech) were incubated with 0.5 µg recombinant FoSir5 protein at 30°C for 1 hr in HDCR buffer (50 mM Tris pH 7.5, 5% glycerol, 5 mM NAD$^+$, 0.1 mM EDTA, 50 mM NaCl, and 0.2 mM PMSF) in the presence of 50, 100, or 200 µM crotonyl-CoA (C4282, Sigma). The assay mixture was then analyzed using Western blotting by anti-PanKcr (PTM-501, PTM Biolabs) and anti-H3 (ab1791, Abcam).

## Immunoprecipitation and mass spectrometry

For identification of FoSir5 interacting proteins, mycelium of Fo and FoSir5-GFP strains were collected and frozen with liquid nitrogen. For total protein extraction, the samples were ground into a fine powder in liquid nitrogen and resuspended in lysis buffer (10 mM Tris-HCl, pH 7.5, 150 mM NaCl, 0.5 mM EDTA, 0.5% NP-40) with 2 mM PMSF and proteinase inhibitor cocktail (Roche). The supernatant lysates were then incubated with anti-GFP agarose (KTSM1301, KT HEALTH) at 4°C for 2 hr with gently shaking. Proteins bound to the beads were eluted after a serious of washing steps by PBS. Elution buffer (200 mM glycine, pH 2.5) and neutralization buffer (1 M Tris base, pH 10.4) were used for the elution process. For identification of crotonylation sites of FoDLAT, total proteins were isolated from FoDLAT-GFP strain and incubated with anti-GFP agarose. The eluted mixture was subsequently analyzed using LC-MS/MS conducted in PTM Biolabs (Hangzhou, China).

## Protein pull-down assays

Coding domain sequence of FoDLAT or FoDLAT$^{K148Q}$ was cloned in pMAL vector for the N-terminal fusion with MBP. The fusion proteins were expressed in BL21 *E. coli*. Transformed cells were induced by adding IPTG to a final concentration of 0.2 mM when OD600 reached 0.6, and the culture was further grown at 37°C for 3 hr. Cells were harvested by centrifugation and lysed by sonication in lysis buffer A. For purification, amylose resin (New England Biolabs) was added to the clarified lysate and incubated for 2 hr at 4°C. Beads were then washed with five column volumes of PBS. MBP fusion proteins were eventually eluted in elution buffer supplemented with 20 mM maltose and then dialyzed at 4°C overnight. For pull-down assay, 1 µg purified FoSir5-His protein was mixed with 1 µg MBP or MBP-FoDLAT protein in the binding buffer (50 mM HEPES, pH 7.5, 1 mM EDTA, 150 mM NaCl, 0.5 mM DTT, and 0.8% glycerol) in a total volume of 100 µl at room temperature for 1 hr; 30 µl of amylose resin was added into the mixture and rotated at room temperature for 1 hr. The mixture was subsequently washed three times with 1 ml of binding buffer, and washed beads were boiled in 30 µl of 2× SDS sampling buffer at 100°C for 5 min. The assay mixture was then analyzed using Western blotting by anti-MBP (New England Biolabs) and anti-His (D2951, Beyotime).

## Generation of anti-K148cr-FoDLAT antibody

FoDLAT K148 site-specific crotonylation antibody was generated by using a FoDLAT crotonylated peptide (KEEKSESK(cr)SESASAC) conjugated to KLH as an antigen. Antibodies were produced from rabbits by HUABIO (Hangzhou, China). The specificity of the antibody was tested by immunoblot analysis.

## In vivo decrotonylation assay

For construction of the RP27:FoDLAT/FoDLAT$^{K148Q}$ /FoDLAT$^{K148R}$:GFP vectors, we amplified fragments by PCR with primers FoDLAT-GFP-F/R, respectively. The fragments were then inserted into the pYF11 vector. Afterward, the constructs were transformed into Fo, ΔFoSir5, and OE-1, respectively. GFP fusion proteins in different pairs of strains were immunoprecipitated as described above. The eluted proteins were then analyzed by Western blot using anti-GFP, anti-Kcr (PTM-501, PTM Biolabs), anti-Ksu (PTM-419, PTM Biolabs), anti-Kma (PTM-902, PTM Biolabs), and anti-Kglu (PTM-1152, PTM Biolabs), followed by quantification using Quantity One (Bio-Rad).

## In vitro decrotonylation assay

Fifty ng of MBP-FoDLAT protein (WT or K148Q) was incubated with or without 50 ng FoSir5-His protein in the absence or presence of 5 mM NAD$^+$ in 200 µl HDCR buffer for 1 hr at 30°C. Samples

were analyzed by Western blot using anti-PanKcr and anti-MBP, followed by quantification using Quantity One (Bio-Rad).

## PDC enzyme assay

PDC activity was measured according to the protocol by PDC activity assay kit (ab109902, Abcam). The germinating conidia of the tested strains were grown in YPD at 25°C for 8 hr in a shaker. The total extracts were diluted and added into the microplate. After incubation in the plate for 3 hr at room temperature, the samples were stabilized and incubated with assay buffer. The fluorescence was measured at 450 nm for 20 min with 20 s interval among each measurement, and the slope of the line indicated the PDC activity. The rates were determined as change in OD over time, represented as change in milliOD per minute.

## Quantification of acetyl-CoA

Acetyl-coA was measured using an acetyl-CoA assay kit (BC0980, Solarbio). The germinating conidia of the tested strains grown in YPD at 25°C for 8 hr in a shaker were harvested and homogenized in lysis buffer of the kit in ice. The supernatant was used to determine acetyl-CoA concentration in triplicate according to manufacturer's instructions.

## Quantification of ATP

The ATP assay kit (S0026, Beyotime), which employs the luciferin-luciferase method (*Drew and Leeuwenburgh, 2003*), was used to quantify ATP. The working solution was prepared according to the kit protocol. The germinating conidia of the tested strains grown in YPD at 25°C for 8 hr in a shaker were harvested and homogenized in lysis buffer. Then, 100 µl of working solution and 20 µl of supernatant of the total extracts were added to each well of a 96-well microtiter plate. The luciferase signals were detected by a multifunctional microplate reader (SpectraMax M2) for 30 s. The standard curve of ATP concentration from 1 pM to 1 µM was prepared by gradient dilution.

## RNA sequencing

The germinating conidia of Fo and ΔFoSir5 with three biological replicates were harvested after growth in YPD medium with shaking at 150 rpm for 8 hr in 25°C. Total RNA was extracted using the TRIzol reagent according to the manufacturer's instructions. RNA-seq data were analyzed as previously described (*Rodenburg et al., 2018*). Briefly, Cutadapt (v1.16) software was used to filter the sequencing data. Reference genome index was built by Bowtie2 (2.2.6) and the filtered reads were mapped to the reference genome using Tophat2 (2.0.14). HTSeq (0.9.1) statistics was used to compare the Read Count values on each gene as the original expression of the gene, and then FPKM was used to standardize the expression. DESeq (1.30.0) was used to analyze the genes of difference expression with screened conditions as follows: an absolute $\log_2$ value >1 and p-value < 0.05. All the detected genes were shown in *Supplementary file 2*.

## Fluorescent qRT-PCR

For qRT-PCR assessment of *FoSir5* expression, fresh spores were inoculated in YPD medium at 25°C with shaking at 150 rpm. At 0, 4, 8, 12, and 24 hr, the cultures were centrifuged at 12,000× *g* for 15 min and the pellets were collected for RNA extraction. For validation of RNA-seq data, three batches of biological repeats of Fo and ΔFoSir5 were independently collected. RNA was extracted and reverse-transcribed using All-In-One RT MasterMix (abm). qRT-PCR was performed using M5 HiPer SYBR Premix EsTaq (Mei5bio). The transcript abundance of candidate genes was calculated using the $2^{-\triangle Ct}$ method, normalized to *FoEF-1α* (elongation factor 1α). All primers used for qRT-qPCR were listed in *Supplementary file 3*.

## ChIP-qPCR analysis

ChIP was performed according to described methods (*Liu et al., 2019*). Briefly, the germinating conidia of different strains were harvested after growth in YPD medium with shaking at 150 rpm for 8 hr in 25°C. The germinating conidia were cross-linked with 1% formaldehyde gently shaking for 25 min and then stopped with glycine with a final concentration of 125 mM for another incubation of 10 min. After cleaning with sterile water for several times, the cultures were frozen and ground with liquid nitrogen.

The powder was re-suspended in the lysis buffer (250 mM HEPES pH 7.5, 1 mM EDTA, 150 mM NaCl, 10 mM DTT, 0.1% deoxycholate, and 1% Triton) and protease inhibitor cocktail (Roche) with a conidia/buffer ratio as 0.2 g/2 ml. The DNA was sheared into ~500 bp fragments using sonicator (Bioruptor Plus CHIP, ultrasonication for 30 s and stop for 30 s, 10 times). The supernatant was diluted after centrifugation with ChIP dilution buffer (1.1% Triton X-100, 16.7 mM Tris-HCl pH 8.0, 1.2 mM EDTA, 167 mM NaCl). Immunoprecipitation was conducted using 5 µl anti-GFP antibody (ab290, Abcam) or 5 µl anti-H3K18cr antibody (PTM-517, PTM Biolabs) together with protein A agarose (Roche) overnight at 4°C. After separation, beads were washed orderly by low-salt wash buffer (150 mM NaCl, 0.2% SDS, 20 mM Tris-HCl pH 8.0, 2 mM EDTA, 0.5% Triton X-100), high-salt wash buffer (500 mM NaCl, 2 mM EDTA, 20 mM Tris-HCl pH 8.0, 0.2% SDS, 0.5% Triton X-100), LiCl wash buffer (0.25 M LiCl, 1% Nonidet P-40, 1% sodium deoxycholate, 1 mM EDTA, 10 mM Tris-HCl pH 8.0), and TE buffer. DNA bound to the beads was then eluted and precipitated. ChIP-qPCR was independently repeated three times. Relative enrichment values were calculated by dividing the immunoprecipitated DNA by the input DNA and internal control gene ($\beta$-tubulin). Primers using for ChIP-qPCR were designed near putative TSS and listed in **Supplementary file 3**.

## Infection assays of *F. oxysporum* on tomato seedlings

Briefly, 2-week-old tomato seedlings were used for root dip infection for 10 min in spore suspension ($10^6$ spores/ml). The infected plants were transplanted in sterile soil-vermiculite mixture (1:1 ratio) and kept in plant growth chamber at 25°C and 90% relative humidity (RH). Severity of disease symptoms was recorded and scored according to the values ranging from 1 to 5: (1) few symptoms, only first true leaf necrotic or curled; (2) clear symptoms, first three leaves developed symptoms; (3) severe symptoms, leaves necrotic and curled, defoliation, growth retardation; (4) rotted plant but still alive; (5) dead plant. Disease index was calculated using the following formula: Disease index=$\Sigma$ (number of leaves in each disease grade × grades value)/(total number of assessed leaves × the highest grade value) (*Yuan et al., 2019*). This inoculation experiment was repeated twice to verify consistency in the observed results. qRT-PCR analysis of *F. oxysporum EF-1α* transcript levels was performed using tomato plants harvested after 14-day infection with different strains. The expression of tomato *RCE1*, a constitutively expressed gene, was used as a control for the use of equal amounts of RNA for RT-PCR.

## Acknowledgements

This research was supported by the National Natural Science Foundation of China (31972213 and 32102149), the Shandong Provincial Natural Science Foundation (ZR2019BC070 and ZR2020KC003), Shandong Province 'Double-Hundred Talent Plan' (WST2018008), and Taishan Scholar Construction Foundation of Shandong Province (tshw20130963). XP and BFL were supported by the Wellcome Trust (200873/Z/16/Z).

## Additional information

### Funding

| Funder | Grant reference number | Author |
| --- | --- | --- |
| National Natural Science Foundation of China | 31972213 | Wenxing Liang |
| National Natural Science Foundation of China | 32102149 | Ning Zhang |
| Shandong Provincial Natural Science Foundation | ZR2019BC070 | Ning Zhang |
| Shandong Provincial Natural Science Foundation | ZR2020KC003 | Wenxing Liang |

| Funder | Grant reference number | Author |
|---|---|---|
| Shandong Province 'Double-Hundred Talen Plan' | WST2018008 | Wenxing Liang |
| Taishan Scholar Construction Foundation of Shandong Province | tshw20130963 | Wenxing Liang |
| Wellcome Trust | 200873/Z/16/Z | Xueyuan Pei<br>Ben F Luisi |

The funders had no role in study design, data collection and interpretation, or the decision to submit the work for publication.

### Author contributions

Ning Zhang, Funding acquisition, Investigation, Methodology, Writing - original draft; Limin Song, Investigation, Methodology, Validation; Yang Xu, Methodology, Validation; Xueyuan Pei, Data curation, Funding acquisition, Software; Ben F Luisi, Funding acquisition, Software, Writing - review and editing; Wenxing Liang, Conceptualization, Funding acquisition, Project administration, Supervision, Writing - review and editing

### Author ORCIDs

Ning Zhang (iD) http://orcid.org/0000-0002-4824-3795
Ben F Luisi (iD) http://orcid.org/0000-0003-1144-9877
Wenxing Liang (iD) http://orcid.org/0000-0002-3791-4901

### Decision letter and Author response

Decision letter https://doi.org/10.7554/eLife.75583.sa1
Author response https://doi.org/10.7554/eLife.75583.sa2

# Additional files

### Supplementary files

- Supplementary file 1. Putative FoSir5 interacting proteins.
- Supplementary file 2. RNA-seq gene expression data.
- Supplementary file 3. Primers used in the study.
- Transparent reporting form

### Data availability

The RNA-seq raw reads are available in NCBI Sequence Read Archive (SRA) database with the accession number of PRJNA687117.

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
