## [Editor Report]

This manuscript is of particular interest to the community of fungal biologists, along with a broader audience studying epigenetic and protein post-translational modifications. The authors combine sophisticated methods to analyze a rather poorly described modification, protein crotonylation mediated by FoSIRT5, and how it affects gene expression/protein activity of metabolically relevant genes/enzymes in Fusarium oxysporum.

---

## [Decision Letter]

**Decision letter after peer review:**

[Editors’ note: the authors submitted for reconsideration following the decision after peer review. What follows is the decision letter after the first round of review.]

Thank you for submitting the paper "The decrotonylase FoSIRT5 facilitates mitochondrial metabolic state switching in conidial germination of *Fusarium oxysporum*" for consideration by *eLife*. Your article has been reviewed by 2 peer reviewers, and the evaluation has been overseen by a Reviewing Editor and a Senior Editor. The reviewers have opted to remain anonymous.

We are sorry to say that, after consultation with the reviewers, we have decided that this work will not be considered further for publication by *eLife*.

This decision was reached based on the consideration that a significant amount of additional work will be necessary to fully address the numerous issues and concerns raised by the reviewers. However, we encourage you to resubmit your manuscript as a new submission if you can resolve these issues experimentally.

*Reviewer #1:*

The present work by Zhang et al. elucidates the fungal deacylase/decrotonylase FoSIRT5 from tomato-infecting Fusarium oxysporum f. sp. lycopersici. Histone/protein crotonylation of lysine residues is a rather recently described and poorly studied modification, therefore, the manuscript presents breakthrough insights into its function in filamentous fungi.

In my opinion, the experimental setup is robust, and the text is well written. It was particularly interesting to see that the authors could link the increase of histone/protein crotonylation by deletion of FoSIRT5 to increased gene expression/protein activity of the acetyl-CoA/ATP biosynthetic machinery, and ultimately to ATP levels required for conidial germination.

Please find my comments below, mainly targeting the fact that little information is provided on (fungal) sirtuin-type deacylases and protein crotonylation in the manuscript, despite it being a rather "novel" modification. Furthermore, the specificity of FoSIRT5 should be addressed by additional experiments.

– The authors describe 5 sirtuin-type deacetylases/deacylases in F. oxysporum (Figure 1A). However, 7 sirtuins have been described in *Neurospora crassa* (Smith et al., 2008, Epigenet Chromatin 1), and indeed, my search with the *N. crassa* sequences revealed two additional hits in F. oxysporum with the Sir2 superfamily protein domain: FOXG_09388 (homolog to *N. crassa* NST-5) and FOXG_12167 (homolog to *N. crassa* NST-7). Please include these two in the manuscript and comment on their predicted localization. I was happy to see that both the Neurospora paper and current work identified NST-6/FoSIRT5 as the closest homologs to human SIRT5.

– Fusarium nomenclature typically consists of three letters and a number (only the first letter capitalized in protein names). Please consider renaming to FoSir5.

– Please comment on other sirtuin-type deacetylases/deacylases from filamentous fungi in the introduction and discussion. E.g., *N. crassa* NST-6 (the FoSIRT5 homolog) was implicated in telomeric silencing (Smith et al., 2008, Epigenet Chromatin 1).

– Please comment on what is known about the availability of crotonyl-CoA in different cellular compartments, on putative crotonyltransferases (e.g., coactivator p300 in human cells; Sabari et al., 2015, Mol Cell 58) and crotonylation reader proteins, and on the putative molecular mechanism of gene activation via histone crotonylation.

– Have additional crotonylation sites been identified in the interacting proteins (Supplementary File 1)?

– Could other histone modifications be removed or other histone lysine residues be targeted by FoSIRT5? Histone acetyl lysine (in particular H3K18ac) and H3K9cr antibodies should be tested.

– Increased and decreased pathogenicity of d-FoSIRT5 and OE-FoSIRT5, respectively, is likely due to increased and decreased germination rates (lines 324-325 and 397).

*Reviewer #2:*

Strengths: the authors present FoSIRT5 decrotonylates a subunit of the fungal pyruvate dehydrogenase complex (FoDLAT) at K148 and histone H3K18 which affects acetyl-CoA levels and ATP contents.

Weaknesses: the authors only show the correlations of FoSIRT5 functions in conidial germination in F. oxysporum, which might be involved with decrotonylating the PDC subunit and H3K18.

The claims are not fully supported by their data.

1) No direct evidence to show FoSIRT5 decrotonylates FoDLAT at K148 by specific antibody or MS/MS data, like the histone H3K18.

2) The author should show whether the crotonylated levels of FoDLAT-K148 alteration during conidial germination in F. oxysporum. Moreover, how is the K148Q/R- FoDLAT affect the germinating conidia?

3) How are the ChIP assays of FoSIRT5 during conidial germination?

4) Figure 4 only presents the correlations of ATP production during conidial germination, with no direct data to introduce the deletion or OE FoSIRT5 during the germination process.

5) The quality of most of the figures is not high, such as low resolutions of fluorescence microscopy data, bar graph figures are lack labeling of the p-value. In figure 2—figure supplement 1. the MS data should highlight the b4 – b3 peaks to reach the Kcr value. In figure D no K148 crotonylation site in the model of FoDLAT.

---

## [Author Response]

[Editors’ note: the authors resubmitted a revised version of the paper for consideration. What follows is the authors’ response to the first round of review.]

Reviewer #1:The present work by Zhang et al. elucidates the fungal deacylase/decrotonylase FoSIRT5 from tomato-infecting Fusarium oxysporum f. sp. lycopersici. Histone/protein crotonylation of lysine residues is a rather recently described and poorly studied modification, therefore, the manuscript presents breakthrough insights into its function in filamentous fungi.In my opinion, the experimental setup is robust, and the text is well written. It was particularly interesting to see that the authors could link the increase of histone/protein crotonylation by deletion of FoSIRT5 to increased gene expression/protein activity of the acetyl-CoA/ATP biosynthetic machinery, and ultimately to ATP levels required for conidial germination.Please find my comments below, mainly targeting the fact that little information is provided on (fungal) sirtuin-type deacylases and protein crotonylation in the manuscript, despite it being a rather "novel" modification. Furthermore, the specificity of FoSIRT5 should be addressed by additional experiments.– The authors describe 5 sirtuin-type deacetylases/deacylases in F. oxysporum (Figure 1A). However, 7 sirtuins have been described in Neurospora crassa (Smith et al., 2008, Epigenet Chromatin 1), and indeed, my search with the *N. crassa* sequences revealed two additional hits in F. oxysporum with the Sir2 superfamily protein domain: FOXG_09388 (homolog to *N. crassa* NST-5) and FOXG_12167 (homolog to *N. crassa* NST-7). Please include these two in the manuscript and comment on their predicted localization. I was happy to see that both the Neurospora paper and current work identified NST-6/FoSIRT5 as the closest homologs to human SIRT5.

We thank the reviewer for the suggestions. The other two *F. oxysporum* Sir2 proteins were now included in Figure 1A. Changes have been made accordingly in the revised manuscript (line 139-147).

2) Fusarium nomenclature typically consists of three letters and a number (only the first letter capitalized in protein names). Please consider renaming to FoSir5.– Please comment on other sirtuin-type deacetylases/deacylases from filamentous fungi in the introduction and discussion. E.g., *N. crassa* NST-6 (the FoSIRT5 homolog) was implicated in telomeric silencing (Smith et al., 2008, Epigenet Chromatin 1).

As suggested, we have changed “FoSIRT5” to “FoSir5” throughout the manuscript. The information on other fungal Sir2 homologues was presented in line 92-101.

– Please comment on what is known about the availability of crotonyl-CoA in different cellular compartments, on putative crotonyltransferases (e.g., coactivator p300 in human cells; Sabari et al., 2015, Mol Cell 58) and crotonylation reader proteins, and on the putative molecular mechanism of gene activation via histone crotonylation.

We now added the comments as suggested in the introduction (line 66-70).

– Have additional crotonylation sites been identified in the interacting proteins (Supplementary File 1)?

We thank the reviewer for raising this point. K148cr was the only Kcr site identified by MS in our analysis. Although some other lysine residues in FoDLAT are also susceptible to crotonylation as indicated by the results, we haven’t found them, probably due to low abundance of these modified sites or technical limitation of the detection.

– Could other histone modifications be removed or other histone lysine residues be targeted by FoSIRT5? Histone acetyl lysine (in particular H3K18ac) and H3K9cr antibodies should be tested.

As reported, SIRT5 has been shown to possess poor deacetylase activity (Du, Zhou et al., 2011). We tested for the modifications with anti-H3K18ac and anti-H3K9cr as suggested, and observe that FoSir5 has little effect on these modifications (Figure 3A, line 250-251).

– Increased and decreased pathogenicity of d-FoSIRT5 and OE-FoSIRT5, respectively, is likely due to increased and decreased germination rates (lines 324-325 and 397).

We agree, and we have changed the text accordingly (line 351-353).

Reviewer #2:Strengths: the authors present FoSIRT5 decrotonylates a subunit of the fungal pyruvate dehydrogenase complex (FoDLAT) at K148 and histone H3K18 which affects acetyl-CoA levels and ATP contents.Weaknesses: the authors only show the correlations of FoSIRT5 functions in conidial germination in F. oxysporum, which might be involved with decrotonylating the PDC subunit and H3K18.The claims are not fully supported by their data.1) No direct evidence to show FoSIRT5 decrotonylates FoDLAT at K148 by specific antibody or MS/MS data, like the histone H3K18.

We are grateful to the reviewer’s constructive comments. To address the reviewer’s concern, FoDLAT K148 site-specific crotonylation antibody (anti-K148cr-FoDLAT) was generated by using a FoDLAT crotonylated peptide (KEEKSESK(cr)SESASAC) as an antigen. The Western analyses with the antibody detected signals with only the wild-type (WT) FoDLAT but not the Q and R mutants, indicating that K148 is indeed crotonylated in vivo (Figure 2C and D). We also found that inactivation and overexpression of FoSir5 led to significantly increased and decreased K148 crotonylation of WT FoDLAT, respectively (Figure 2C and D). Besides, FoSir5 also plays a role in the decrotonylation of FoDLAT K148cr in vitro (Figure 2E). These data demonstrate that FoSir5 is responsible for K148 decrotonylation of FoDLAT. Changes have been made accordingly in the revised manuscript (line 209-222, 579-583).

2) The author should show whether the crotonylated levels of FoDLAT-K148 alteration during conidial germination in F. oxysporum. Moreover, how is the K148Q/R- FoDLAT affect the germinating conidia?

We thank the reviewer for the concerns. (1) Based on the suggestions, we immunoprecipitated FoDLAT-GFP proteins at different time points during germination and detected the crotonylation level of K148 using anti-K148cr-FoDLAT. As shown in Figure 4A, the K148cr of FoDLAT exhibited a rapid growth at 8 hpr and kept a relatively high level afterwards (12-24 hpi). (2) We have already demonstrated in Figure 5E and F that K148Q mutation of FoDLAT led to elevated ATP and conidial germination, whereas the K148R mutant strain demonstrated decreased ATP level and germination.

3) How are the ChIP assays of FoSIRT5 during conidial germination?

We thank the reviewer for the query. We have demonstrated that FoSir5 is responsible for decroronylation of H3K18cr. During conidial germination, (1) *FoSir5* expression decreased; (2) the level of H3K18cr was elevated; (3) the target genes directly regulated by FoSir5 and H3K18cr showed increased expression. Thus, we speculated that declined enrichment of FoSir5 in these genes happened during germination. As suggested by the reviewer, we performed the ChIP assays during conidial germination using anti-GFP antibody in FoSir5-GFP strain driven by the native promoter. The results illustrated that declined enrichment of FoSir5 in promoter regions of 8 energy-related genes was observed during germination (Figure 4F, line 293-294).

4) Figure 4 only presents the correlations of ATP production during conidial germination, with no direct data to introduce the deletion or OE FoSIRT5 during the germination process.

We thank the reviewer for this point. As suggested, we have now measured the ATP content in FoSir5 mutant and overexpression strains. The results show that ΔFoSir5 and OE-1 exhibit a continuous high and low level of ATP during the whole germinating process, respectively (Figure 4—figure supplement 1), further confirming the relationship between FoSir5 and ATP. Changes have been made to the text (line 298-301).

5) The quality of most of the figures is not high, such as low resolutions of fluorescence microscopy data, bar graph figures are lack labeling of the p-value. In figure 2—figure supplement 1. the MS data should highlight the b4 – b3 peaks to reach the Kcr value. In figure D no K148 crotonylation site in the model of FoDLAT.

Thanks for pointing this mistake out. It has been corrected in the new figures with higher resolution.

As presented in many related articles (Bulgarelli, Rott et al., 2012, Chen, Wang et al., 2018, Nützmann, Reyes-Dominguez et al., 2011, Sarikhani, Mishra et al., 2018), we perform statistical analysis as follows: the different letters above the mean values of three replicates indicates a significant difference between different samples (p < 0.05, ANOVA) and the double asterisks represent p < 0.05 (t-test). We can provide the p value if needed.

Upon the reviewer’s request, we highlighted the b3 and b4 peaks in new Figure2—figure supplement 1B.

The model of the FoDLAT domains were predicted based on homology templates in human and yeast. Unfortunately, crystal structure of FoDLAT between aa. 126-183 was missing, probably due to the lack of filamentous fungi templates information. As an alternative, we marked this area with a dotted line and pentacle indicates the K148 crotonylation site in new Figure2—figure supplement 1C.

References:

Bulgarelli D, Rott M, Schlaeppi K, Ver Loren van Themaat E, Ahmadinejad N, Assenza F, Rauf P, Huettel B, Reinhardt R, Schmelzer E, Peplies J, Gloeckner FO, Amann R, Eickhorst T, Schulze-Lefert P (2012) Revealing structure and assembly cues for Arabidopsis root-inhabiting bacterial microbiota. *Nature* 488: 91-5

Chen Y, Wang J, Yang N, Wen Z, Sun X, Chai Y, Ma Z (2018) Wheat microbiome bacteria can reduce virulence of a plant pathogenic fungus by altering histone acetylation. *Nat Commun* 9: 3429

Du J, Zhou Y, Su X, Yu JJ, Khan S, Jiang H, Kim J, Woo J, Kim JH, Choi BH, He B, Chen W, Zhang S, Cerione RA, Auwerx J, Hao Q, Lin H (2011) Sirt5 is a NAD-dependent protein lysine demalonylase and desuccinylase. *Science* 334: 806-9

Nützmann HW, Reyes-Dominguez Y, Scherlach K, Schroeckh V, Horn F, Gacek A, Schümann J, Hertweck C, Strauss J, Brakhage AA (2011) Bacteria-induced natural product formation in the fungus Aspergillus nidulans requires Saga/Ada-mediated histone acetylation. *Proc Natl Acad Sci U S A* 108: 14282-7

Sarikhani M, Mishra S, Maity S, Kotyada C, Wolfgeher D, Gupta MP, Singh M, Sundaresan NR (2018) SIRT2 deacetylase regulates the activity of GSK3 isoforms independent of inhibitory phosphorylation. *eLife* 7